# The Increasing Coastal Urbanization in the Mediterranean Environment: The State of the Art in Italy

Daniela Smiraglia , Alice Cavalli * , Chiara Giuliani and Francesca Assennato

Italian Institute for Environmental Protection and Research (ISPRA), Via Vitaliano Brancati 48,
I-00144 Rome, Italy; daniela.smiraglia@isprambiente.it (D.S.); chiara.giuliani@isprambiente.it (C.G.);
francesca.assennato@isprambiente.it (F.A.)
* Correspondence: alice.cavalli26@gmail.com

**Abstract:** This paper describes the state of the art of urbanization in Italian coastal areas in 2021, both at national and regional level. Moreover, we focused on six coastal municipalities, aiming to evaluate land consumption in relation to population dynamics between 2012 and 2021 and assessing per capita consumed land in each municipality. Finally, an analysis of land consumption in specific areas prone to natural risks along the coastline (hydraulic, landslide and seismic, hazard) was provided. We considered areas of medium hydraulic hazard, of high and very high landslide hazard, and of high and very high seismic hazard. The results indicate an intense process of urbanization in the first 1000 m from the coastline at national, regional, and municipal levels, which is also increasing in the presence of stabilization or dwindling inhabitants. Furthermore, urbanization is also affected by geomorphology, leading to the developments of settlements in the most accessible areas, such as coastal plains, without taking into consideration the presence of natural hazards. The study highlights the importance of monitoring land consumption to the understanding of processes related to urbanization in coastal areas, from the perspective of future effective policies and to support sustainable planning.

**Keywords:** land monitoring; soil sealing; land degradation; hydrogeological hazard; land consumption





## 1. Introduction

The loss of natural and productive land from urbanization is a global phenomenon, and represents one of the most important factors in landscape change and land degradation [1–3], associated with the loss of ecosystem services [4–6] and an increasing desertification risk [7,8]. Urbanization has been recently referred to within the concept of land consumption, considered as the change from non-artificial land cover to artificial land cover, with a distinction having to be made between permanent consumption and non-permanent consumption [9]. It is widely diffused in both compact and dispersed patterns [10,11]. Urban densification in the consolidated city as well as sprawling phenomena in fringe and rural areas have become a matter of intense investigation [9,12–16]. Furthermore, the need for an accurate analysis and regulation in the European context is becoming increasingly urgent for the formulation of efficient territorial policies [17,18].

The new EU soil strategy for 2030 [19] is a key deliverable of the EU biodiversity strategy for 2030 [20]. It aims to pursue Green Deal objectives [21] and to reinforce the importance of soils to tackle the challenges of climate change, desertification, land degradation, and biodiversity loss, and to guarantee many ecosystem services. Furthermore, it considers artificialization as one of the main topics to be included in the forthcoming EU Soil Health Law. The vision of the new strategy is to have all EU soil ecosystems healthy and more resilient by 2050, defining medium and long-term goals. With reference to land take and soil sealing, i.e., the expansion of cities and infrastructures at the expense of agriculture and natural environment [1], the strategy includes several actions to set ambitious national,

regional, and local targets by 2023, to reduce net land take by 2030 and to achieve net zero soil consumption by 2050. Among them, to integrate the "land take hierarchy", which gives absolute priority to the reuse of already built and sealed areas, into municipal plans. This may help to avoid new construction and sealing on vegetated or permeable soils, thereby protecting soils through appropriate regulatory initiatives and the phasing out of financial incentives contrary to this hierarchy.

Starting from land take and soil sealing definitions in soil health legislation, the European Commission intends to propose a definition of consumption. It would allow different options to be considered for monitoring and reporting progress towards net zero consumption targets. This is also in view of the implementation of a land consumption hierarchy based on data reported by Member States, and to provide guidance to public authorities and private companies on how to reduce soil sealing.

The strategy highlights the importance of the concept of soil health, which refers to soil as a living organism to be maintained in good health to ensure the health of all of us, defining soils as healthy when they are in good chemical, biological and physical condition and can therefore continuously provide for as many ecosystem services as possible.

Although general soil protection legislation is not in place in Italy, with the new National Recovery and Resilience Plan, the Italian Government has committed to approving a "national law on land consumption in compliance with European objectives, which declares the fundamental principles of reuse, urban regeneration and limitation of its consumption". This may pave the way to planning a series of interventions and investments focused on protecting the environment by promoting the ecological transition, preserving natural resources, securing hydrogeological risk areas, and limiting land consumption. Moreover, Italy now anticipates the achievement of net zero land consumption by 2030 instead of 2050, including the target in the National Plan for Ecological Transition approved in 2022.

Over the last century, Europe has undergone important economic, social, and cultural changes [14,22,23], leading to the abandonment of hilly mountain areas and the expansion of urban and industrial settlements in flat areas [24–26]. In particular, southern European cities experienced population concentration increases up to the 1980s, before a moderate deconcentration [27]. The strong human pressure in those areas led to major expansion of construction activities, mainly around the major cities and along the seacoast [28–30], where urban extent is even faster than in other regions [31]. Most of the Mediterranean population is concentrated in the coastal region [32,33]. Housing (mostly secondary housing in many areas), services and tourism are the main factors of coastal land uptake using artificial surfaces, coupled with mostly deregulated urban growth [22,30].

In the Mediterranean basin, unplanned (or poorly regulated) settlement expansion is typical of several coastal urban regions [18,34,35]. Artificial surfaces increase in natural coastal areas, in many cases faster than population increase [13,18,22]. The processes of littoralization and overexploitation of natural resources strongly affect the sensitivity to degradation of these vulnerable ecosystems [36,37]. Furthermore, global climate change amplifies this phenomenon, accelerating coastal erosion, habitat loss, and soil and groundwater salinization processes [34,38–40]. Coastal areas are expected to experience a 160% increase in urban extent between 2000 and 2030 [31], and many cities will potentially be exposed to biophysical hazards [41,42].

In Italy, coastal landscapes amount to almost 8300 km, and their biodiversity and cultural heritage are a distinctive part of Italian territory, which is often subject to land consumption. Artificial surfaces in coastal areas are mainly due to industrial, defense and harbour constructions, and touristic exploitation; every year, natural and agricultural typologies are replaced by over 10 km of anthropic cover (source: https://sinacloud.isprambiente.it/portal/apps/sites/#/coste (accessed on 31 January 2023)). Moreover, despite the 300 m strip of coastal land protected by national law (d.lgs 42/2004), uncontrolled coastal urban development due to the lack of policy and governance integration and coordination has led to unsustainable overexploitation of fragile ecosystems, triggering

phenomena of land degradation [11,35,40] and increasing the exposure of the population and ecosystems to hazards. Within this context, accounting for spatiotemporal urban land cover changes and biophysical hazards is necessary to monitor coastal areas and to guide policies in future urban development [43,44].

On the basis of land consumption data at a national level [32] which is published every year by the Italian Institute for Environmental Protection and Research (ISPRA), in this paper, we present the state of the art of coastal land consumption in Italy, highlighting the situation even in areas at hydrogeological risk, affecting the vulnerability of ecosystems. The analysis was performed at national and regional levels for the year 2021 in three coastal strips along a gradient from coast to inland. Besides the immediate strip of coastal land protected by national law (0–300 m), assessment of coastal zone is enhanced by also investigating the strip up to 1 km (300–1000 m) and the coastal hinterland (1000–10,000 m), both according to European monitoring programs such as the European Environment Agency [22] and the Copernicus Land Monitoring Service (CLMS) local component "Costal Zones" (https://land.copernicus.eu/local/coastal-zones (accessed on 15 January 2023)).

Furthermore, a focus was placed on six coastal municipalities, chosen among those with the highest percentages of consumed land recorded in the first coastal strip (0–300 m), aiming to (i) assess land consumption in the three coastal strips; (ii) assess land consumption trends in the period 2012–2021, also in relation to population dynamics; and (iii) assess land consumption extension in specific areas prone to hydraulic, landslide, and seismic hazards.

After having introduced the topic of urban expansion in coastal landscapes, we describe the input data and methodology used to assess coastal land consumption in Italy, also in relation to population dynamics and hydrogeological hazards. We then present the results and discuss the main implications of the analysis. The final section proposes some concluding remarks derived accordingly.

## 2. Materials and Methods

### 2.1. Study Area

The study area is the coastal zone of the Italian territory (Figure 1), specifically from the coastline up to 10 km. The total area is 50,041 km$^2$, and includes the six municipalities of the study (1319 km$^2$). The climate is Mediterranean, with dry summers and mild winters. Land cover is characterized by forests, grasslands, built-up areas, and wetlands. Herbaceous vegetation is dominated by cropland and pastures and is mainly concentrated on coastal areas and plains. The orography affects the distribution of the settlements, which are concentrated in flat and coastal areas. Land consumption along coastal zones is more widespread in low and sandy shores, even if they represent the 41% of the total coastline, while high and rocky shores cover 59% of the total [33]. The six municipalities analysed in this study are in the north of the peninsula (Venice, Genova, and Livorno) and in the south (Bari, Taranto, and Messina) (Figure 1).

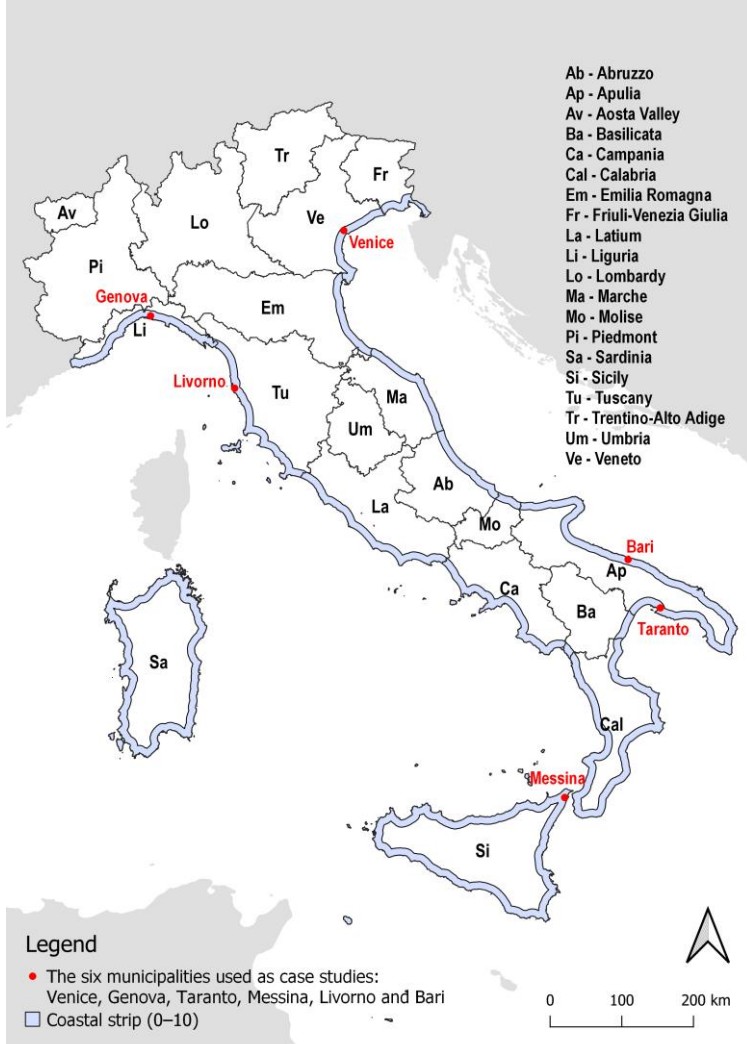

**Figure 1.** Coastal zone of the Italian territory (up to 10 km from the coastline). The map shows the boundaries of the 20 administrative regions and the location of the six municipalities.

*2.2. Reference Data*

The analysis of land consumption in the three coastal strips and in the six municipalities is based on the national land consumption map of Italy (available at: https://groupware.sinanet.isprambiente.it/uso-copertura-e-consumo-di-suolo/library/consumo-di-suolo (accessed on 10 January 2023)), produced every year at 10 m spatial resolution, and mapped both in its permanent and temporary forms. On the basis of the national methodology [45], the national Land Consumption Map initially generated using RapidEye imagery for the year 2012 is updated every year, given the data availability from 2015 of Sentinel-1 and Sentinel-2 [9]. The manual photointerpretation is carried out on a detailed scale (greater than 1:5000, MMU 100 $m^2$) based on Sentinel-2 mosaics, national orthophotos and other free VHR satellite images. The data are converted into raster format (10 × 10 m spatial resolution) by assigning each pixel its class according to a third-level hierarchical classification that distinguishes consumed and non-consumed and reversible/irreversible land consumption, for the distinction of land cover [9]. For the purpose of assessing the coastal land consumption in Italy, the aforementioned classification system was condensed into two basic categories: consumed and non-consumed land (Figure 2).

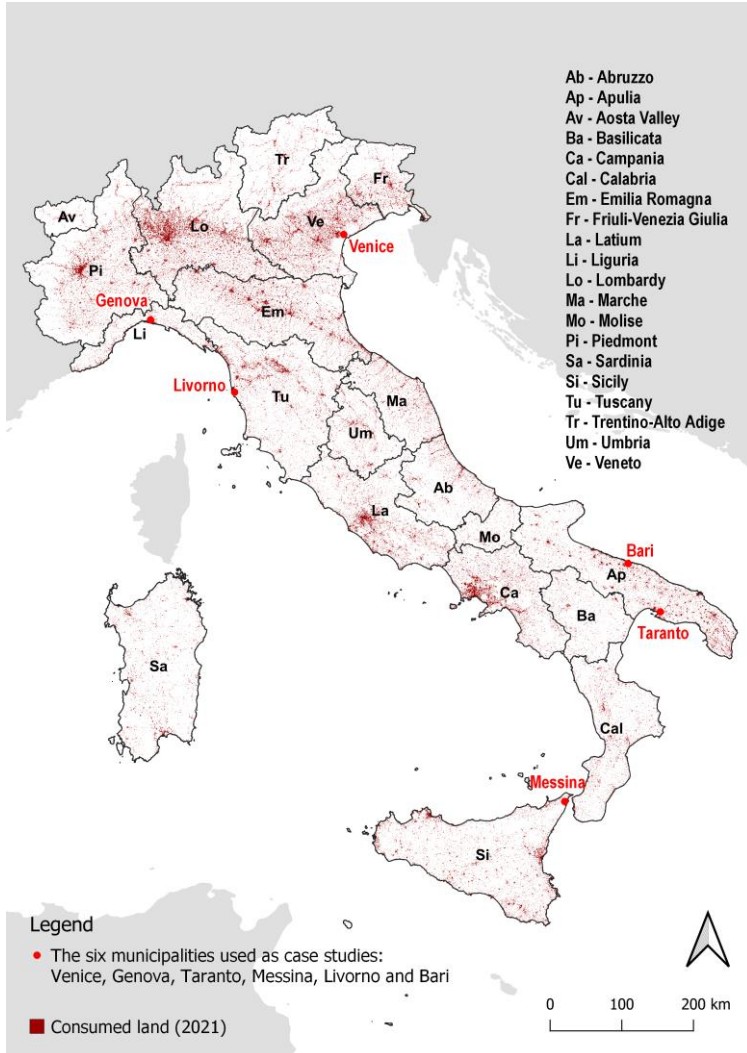

**Figure 2.** Land consumption map of Italy (year 2021).

The analysis refers also to data from the Coastline Geodatabase from ISPRA, which includes the digitisation of the entire national coastline, defence works, harbour works, the backshore boundary with associated land use characterisation, and the delimitation of the areas of beaches, made available for the years 2000, 2006 and 2020 (https://sinacloud. isprambiente.it/portal/apps/webappviewer/index.html?id=089e0739893f482e9e9b62736 0b6ff6d (accessed on 10 January 2023)). The national coastline was used as a reference to generate the coastal area from the coastline up to 10 km (Figure 1), which was then divided into three coastal strips (0–300, 300–1000, 1000–10,000 m).

The landslide and hydraulic hazard maps (Figure 3) are derived from the new national mosaics [34] made on the basis of the Hydrogeological Structure Plans-PAI Landslides (v.4.0-2020–2021) and the hydraulic hazard maps drawn up by the District Basin Authorities, according to the scenarios of Legislative Decree 49/2010, implementing the Floods Directive (2007/60/EC), which aims to create a homogeneous framework at European scale for the management of flooding phenomena in order to reduce risks. For seismic hazard areas, ISPRA data are integrated with the reference data of the National Institute of Geophysics and Volcanology [46,47].

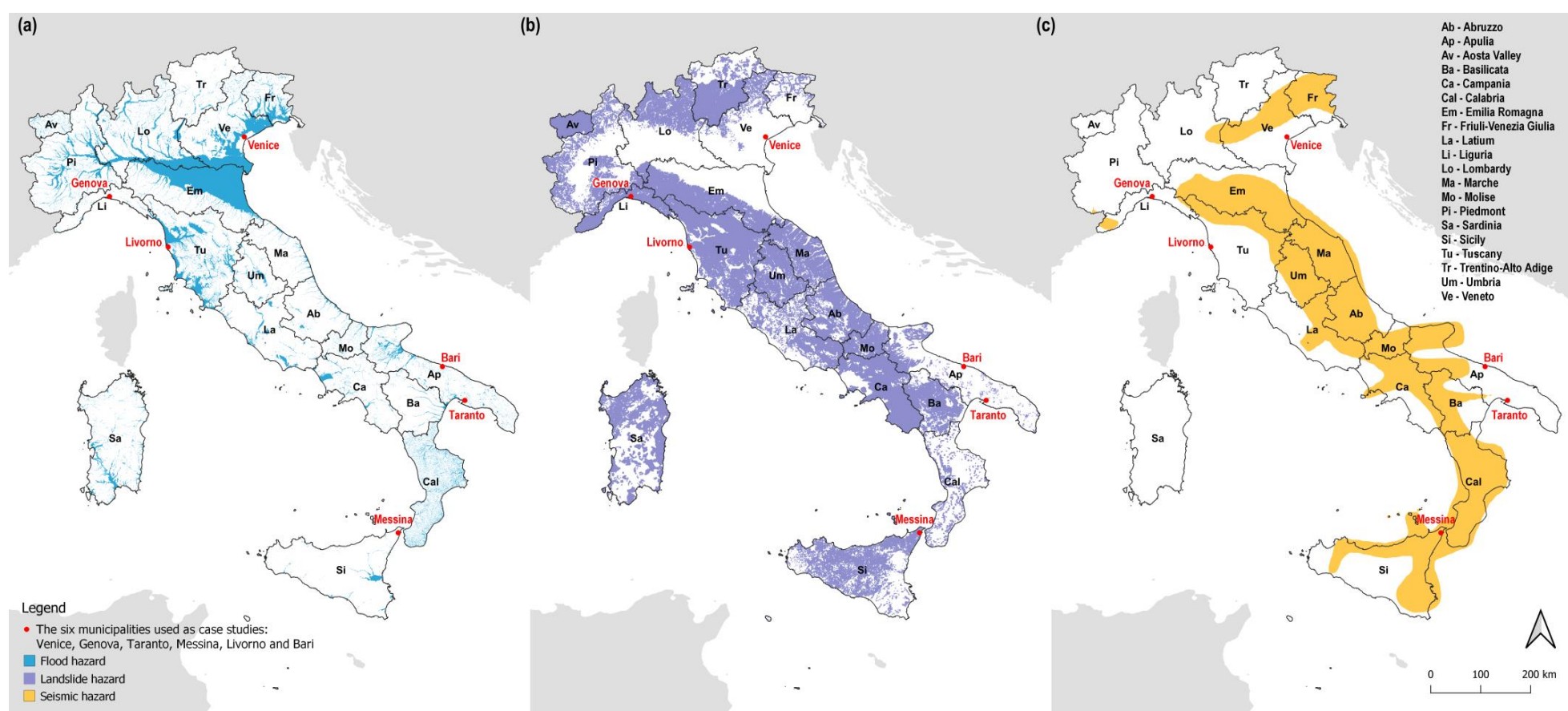

**Figure 3.** Maps of hydraulic (**a**), landslide (**b**), and seismic (**c**) hazard areas of Italy.

The six municipalities used as case studies (Venice, Genova, Taranto, Messina, Livorno and Bari) were chosen based on the highest percentages of consumed land recorded in the first coastal strip (0–300 m) (Figure 4). Resident population data of the six municipalities were derived from the population database published by the Italian National Institute of Statistics (ISTAT), and were gathered for the years 2012 and 2021 (http://dati.istat.it/Index.aspx?DataSetCode=DCIS_POPRES1 (accessed on 5 January 2023)).

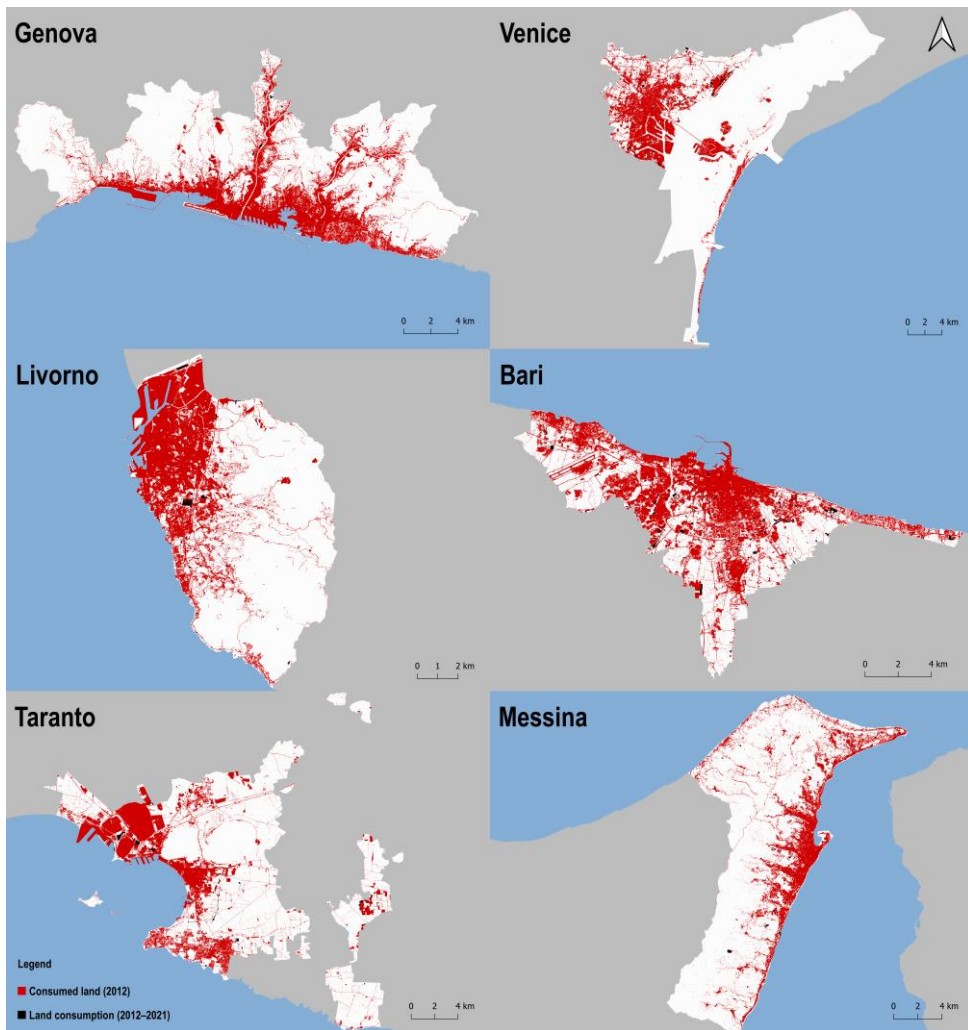

**Figure 4.** Land consumption maps of the six municipalities. Black represents the consumed land in 2012, and red the changes that occurred between 2012 and 2021.

### 2.3. Assessment of Coastal Areas Changes

The coastal area map, divided into three coastal strips (0–300, 300–1000, 1000–10,000 m), was overlaid on the national land consumption map, which was grouped into consumed and non-consumed land, to analyse the distribution of land consumption in the three coastal strips for the national coastal area, both for the administrative regions and for the six municipalities selected.

After the assessment of consumed land in the three coastal strips at national, regional, and municipal level, the analyses also covered the assessment of land consumption trends from 2012 to 2021 in the six municipalities selected, reported as percentage of change against the value of consumed land in 2012. Resident population data from ISTAT database were used to assess per capita consumed land in each municipality. The land consumption trend between 2012 and 2021 was calculated considering the normalized values of land consumption that occurred between 2012 and 2021 (year 2012 = 100). The assessment of

per capita consumed land was calculated using the ratio between consumed land and the resident population (m$^2$/p), both for 2012 and 2021.

Furthermore, an analysis of coastal hydrogeological risk was made using the overlay between the land consumption map and the hydraulic, landslide, and seismic hazard maps in the six municipalities. Regarding the three types of natural hazard, we considered areas of low, medium, high and very high hydraulic and landslide hazard, and of high and very high seismic hazard. The surface of land consumed by each type of hazard was calculated. The analyses were performed using QGIS 3.22 software.

### 3. Results

Figure 5 shows coastal land consumption in Italy, and Table 1 shows consumed land on a national level, in hectares and percentages, in the three coastal strips for the year 2021, amounting to 63,206 ha (22.47%), 87,848 ha (19%), and 369,462 ha (8.67%), respectively. Although the third coastal strip recorded the highest value of consumed land, the percentages over the strip extension reveal how the first and second coastal strips are the ones most involved in urbanization processes.

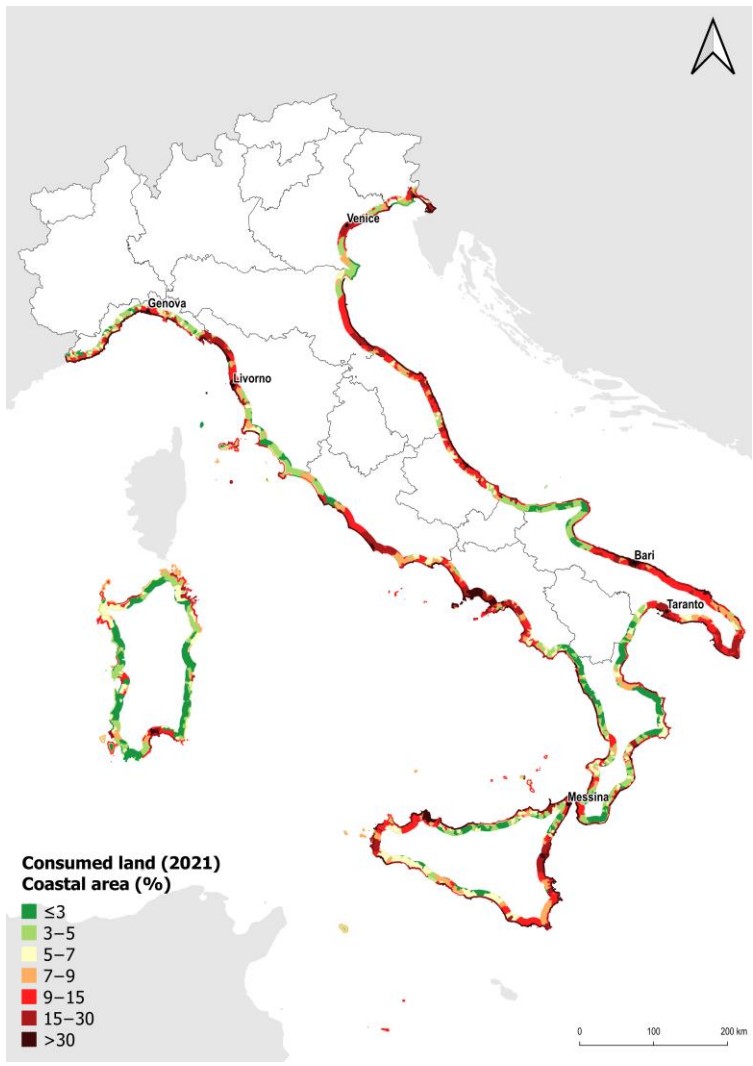

**Figure 5.** Map of coastal land consumption in Italy [45].

**Table 1.** National consumed land, in hectares and percentages, in the three coastal strips for the year 2021.

| Distance from the Coastline (m) | Consumed Land 2021 (ha) | Consumed Land 2021 (%) |
|---|---|---|
| 0–300 | 63,206 | 22.47 |
| 300–1000 | 87,848 | 19.00 |
| 1000–10,000 | 369,462 | 8.67 |

Figure 6a sums up the consumed land assessment at a regional level in the three coastal strips. The regions most affected are Campania (18.67%), Emilia-Romagna (14.06%), and Marche (14.55%), whereas those less affected are Basilicata (3.98%) and Sardinia (5.14%). Regarding the three coastal strips (Figure 6b), the Liguria and Marche regions are the most affected by consumed land in the first coastal strip (0–300 m) (47.02% and 45.07%, respectively), followed by Abruzzo, Emilia-Romagna, and Campania (between 35% and 37% of consumed land). Emilia-Romagna, Abruzzo, and Campania are also the regions most affected by consumed land in the second coastal strip (300–1000 m) (34.81%, 32.62%, and 30%, respectively), and Campania is the region with the highest percentage of consumed land in the third coastal strip (1000–10,000 m). Basilicata and Sardinia show the lowest percentages (between 2% and 8%) in the three areas.

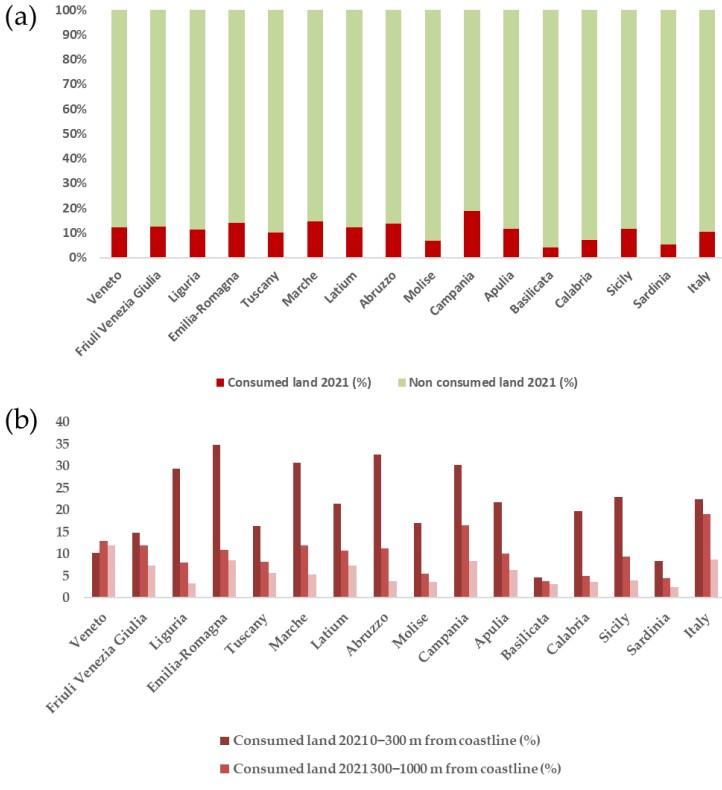

**Figure 6.** Sum of the percentage of consumed land by 2021 of the three coastal strips for each administrative region (**a**), and percentage of consumed land by 2021 in the coastal strips separately for each administrative region (**b**).

Regarding the assessment of consumed land in the six municipalities, Figure 7a shows Bari as the municipality most affected by consumed land (43%) followed by Livorno (28%) and Genova (24%). As shown in Figure 7b, these three municipalities are also the ones with the highest percentages of consumed land in the first (Genova 78%, Bari 68%, and Livorno 63%) and second coastal strips (Bari 58%, Genova 58%, and Livorno 48%). In the third

coastal strip Bari, Venice, and Taranto are the regions most affected by consumed land with 38%, 25%, and 18% of their land being consumed.

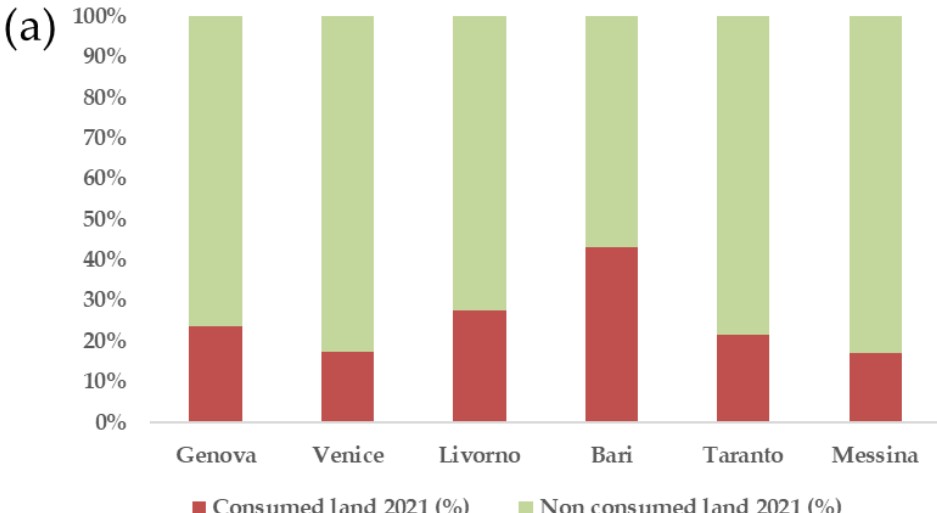

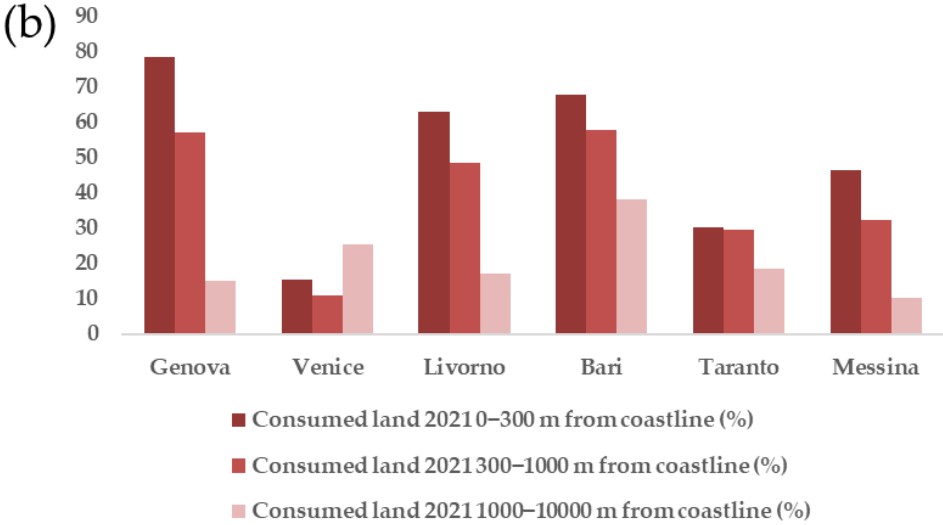

**Figure 7.** Sum of the percentage of consumed land by 2021 for the six municipalities (**a**), and percentage of consumed land by 2021 in the coastal strips separately for the six municipalities (**b**).

Analysing the trend of the phenomenon and using the yearly increase in consumed land over the period 2012–2021 as an indicator (Figure 8a), the Bari and Genova municipalities recorded the highest and the lowest trend increases, respectively, whereas Messina, Taranto and Bari's land consumption increased considerably between 2012 and 2015, and Bari and Venice's between 2017 and 2019. Figure 8b shows the relationship between consumed land and population dynamics. All the six municipalities considered in this analysis show a demographic decrease between 2012 and 2021: Genova and Messina decreased by 28,802 and 20,330 inhabitants, respectively, whereas Venice, Taranto, Livorno, and Bari decreased by less than 10,000 inhabitants. The greater increase in consumed land per capita occurred in Venice, Taranto, and Messina, where in 2012 it was 264 $m^2$/p, 260 $m^2$/p and 147 $m^2$/p, respectively, while in 2021 it reached 280 $m^2$/p for Venice and Taranto and 165 $m^2$/p for Messina.

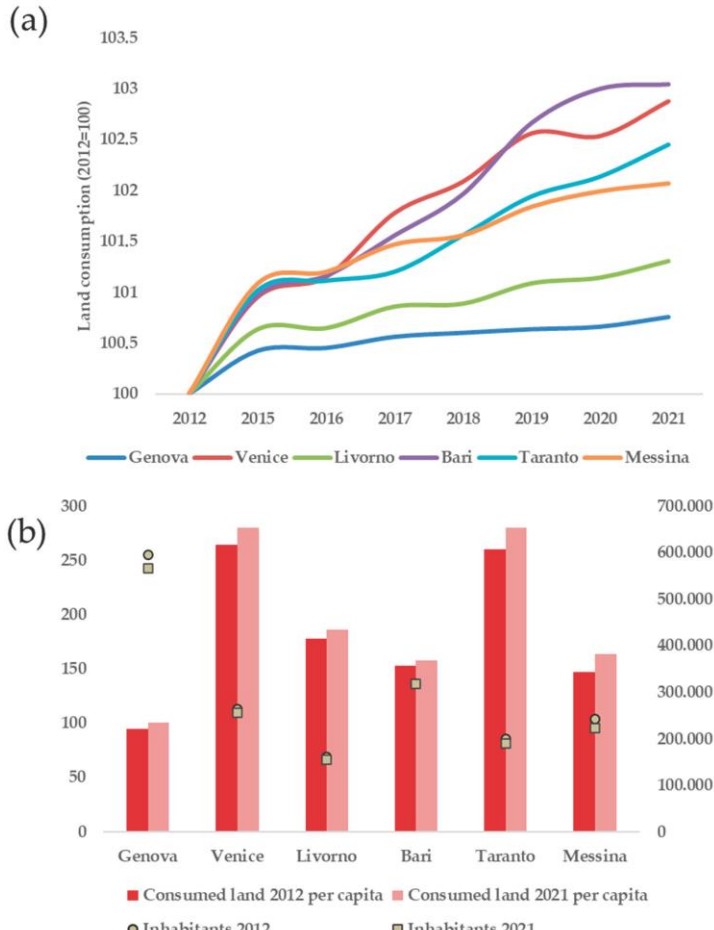

**Figure 8.** Trend of land consumption increases (2012–2021) normalized to 100 (2012 = 100) of the six municipalities (**a**), and the relationship between consumed land and population dynamics in 2012 and 2021 (**b**).

Figure 9 shows the distribution of land consumption in Italy in coastal areas of moderate, medium, high and very high hydraulic hazard (Figure 9a), in high and very high landslide hazard areas (Figure 9b), and in areas of high and very high seismic hazard (Figure 9c). Figure 10 shows the results of the assessment in the six municipalities of consumed land in hydrogeological risk areas. All the municipalities are affected by consumed land in hydraulic hazard areas, with the greatest percentage occurring in Genova (68%), followed by Livorno (51%) and Venice (40%). As for landslide hazard, Bari recorded the highest percentage (37%), and lower percentages were recorded in Messina (7%) and Livorno (5%). Eventually, the Messina municipality is the only one in which consumed land is present in seismic hazard areas, and is the one in which consumed land is present in all three types of hazard areas.

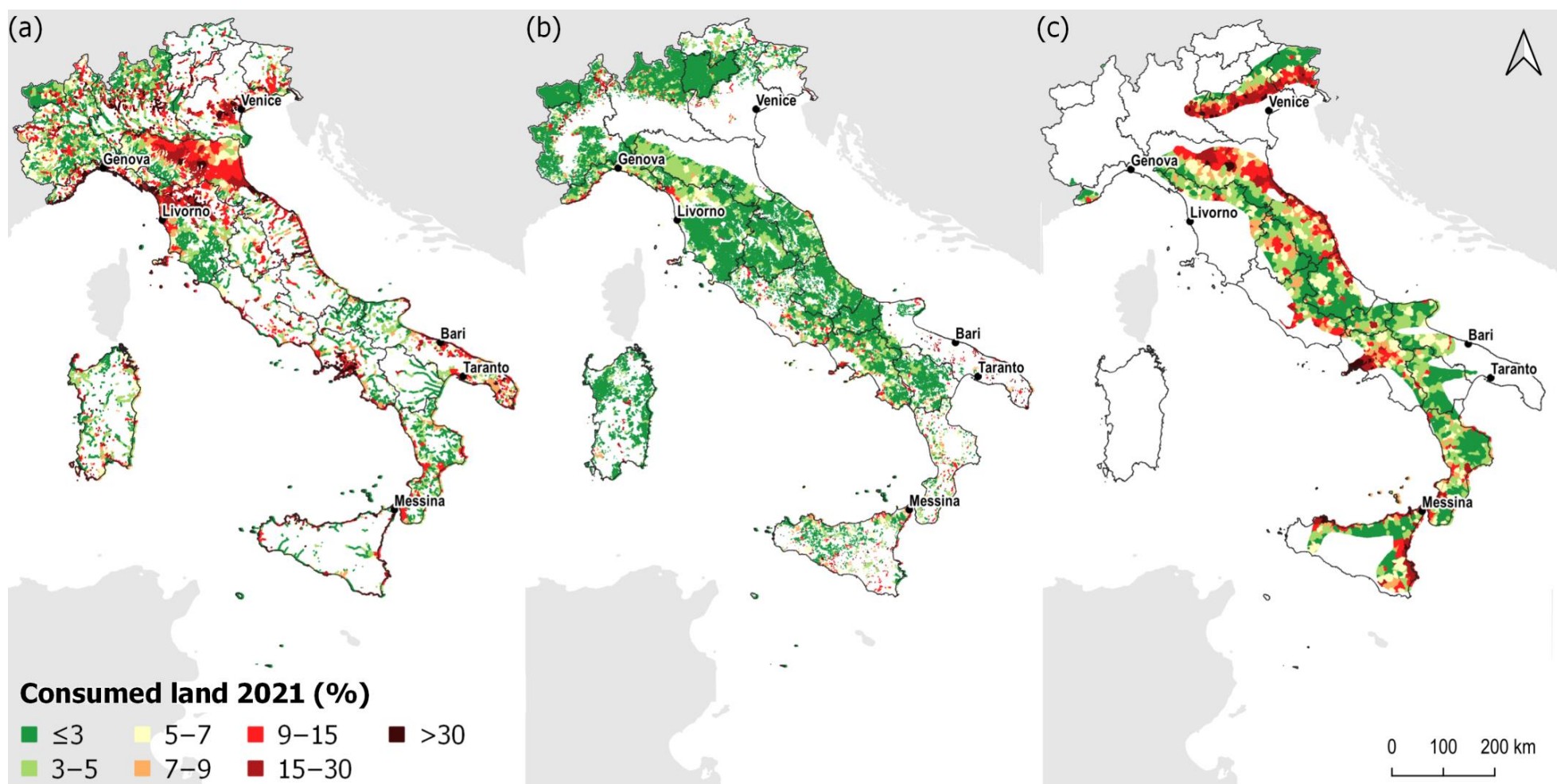

**Figure 9.** Land consumption (%) in coastal areas of low, medium, high, and very high hydraulic hazard areas (**a**), low, medium, high, and very high landslide hazard areas (**b**), and high and very high seismic hazard areas (**c**) of Italy [45].

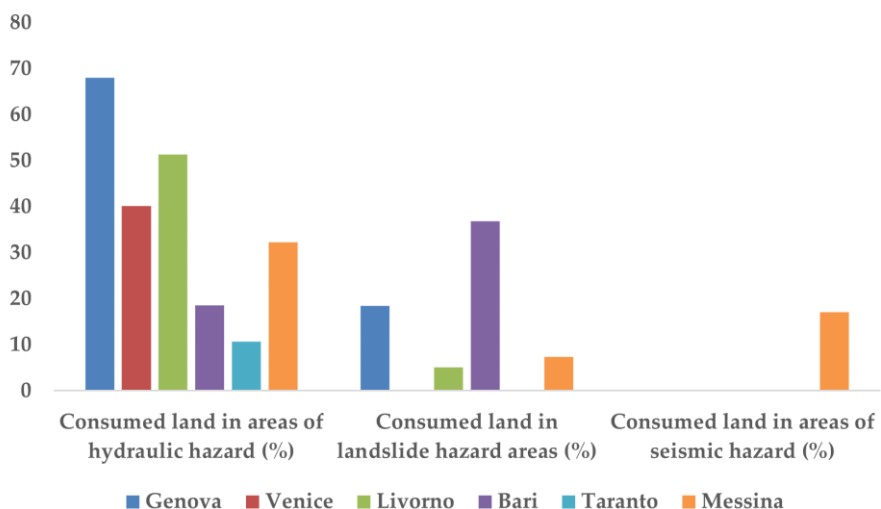

**Figure 10.** Distribution of consumed land in hydrogeological risk areas for the six municipalities (year 2021).

## 4. Discussion

This study draws a picture of the Italian coastal areas that after World War II underwent an intense process of uncontrolled urbanization, especially in particular regions [36]. At the national level, the first 1000 m from the coastline are the areas to pay attention to, because these are the areas most exploited for human activities [11] and the ones in which the greatest land consumption is recorded. This is also shown at regional and municipal level, especially in the Liguria region, which even if it does not show the highest percentage of artificial surfaces in the coastal area, has the highest percentage in the first strip and so also in the municipality of Genova. Only the Veneto region has a higher percentage in the third strip, probably due to regional coast geomorphology, with a wide extent of lagoons; this situation is also found in the municipality of Venice. The high values of consumed land within the three coastal strips, which are certainly affected also by the geomorphology given the presence of wide coastal plains in many Italian regions, confirm the need to stop land consumption, especially in these fragile ecosystems in which natural resource exploitation has already occurred.

Other relevant information can be gleaned from the trend of urbanization, which highlights how in some municipalities such as Venice, Taranto and Bari, urbanization has not slowed down; additionally, in other municipalities, it is still increasing, even in the presence of stabilization or dwindling inhabitants. Indeed, the increase in consumed land per capita shows the imbalance of land consumption when compared to demographic trends. This is also confirmed in other studies of European cities which have shown that even if the population decreases, the urban extent increases, driven more by real estate speculation than by net population growth [30,31,48,49].

The results of the analyses at municipal level show how consumed land within the areas with hydraulic, landslide, and seismic hazards contributes to an increase in the risk degree of the territory. Indeed, each municipality is subject to at least one of the hazards considered in the study, occasionally with high percentages. For example, the municipalities of Genova and Livorno are to be taken into consideration for the high percentages of consumed land in hydraulic hazard areas, and Bari for the high percentage in the landslide hazard areas, but above all, the municipality of Messina, since the consumed land is present in all three types of hazard areas. In Europe, the hazards considered in this study are responsible for the largest economic losses, but floods and storms are the most frequently registered events [50]. For this reason, the consumed land in areas of hydraulic hazard, which involves all the municipalities in this study, is a key piece of information to take into account, considering the impact of waterproofing and the consequent surface runoff. These results demonstrate just how problematic additional consumption of land in hazard-prone areas is, worsening

the impacts of extreme events such as earthquakes, landslides or flooding can have on people, goods, and services. In Italy, many villages are located on or at the bottom of cliff coastal areas, and are threatened by rock falls and slides that can be triggered by heavy storms or by earthquakes, especially along the Campania, Calabria and Sicily coasts [51]. The number of inhabitants can increase 10 times compared to in winter, and the overall risk in coastal areas is very high. Therefore, appropriate studies and countermeasures are needed.

This study presents some examples of municipal scale assessment, and could be improved for an extensive evaluation of all Italian coastal areas by adding statistical analysis to better explore the evolution of land consumption, with particular reference to population dynamics. More detailed studies are also needed on the development of land consumption and its relationship to coastal hazards such as landslides, an area that is has garnered increasing attention in recent years [52–54], particularly in Italy, after devastating events in southern areas (e.g., Ischia island in 2022 and the Amalfi coast in 2010). Monitoring activities such as the one presented in this study can be used efficiently at regional and river basin levels to inform policy-makers about the risks of intensification of urbanization in coastal areas, supporting national policies on the integrated management of biodiversity, water, and soil. Furthermore, practical applications in land use planning can be used to avoid urban intensification in densely populated or risky areas. Great effort is needed to achieve efficient land planning and management, in view also of global changes that may intensify the negative effects of the aforementioned phenomena.

## 5. Conclusions

In this study, regional and municipal levels were chosen as examples to address the troubling issues of coastal urbanization. Although in Italy, this process had been more evident until the 80s [18] (with high rates of change), this process is still ongoing, even in areas at hydrogeological risk, affecting the vulnerability of ecosystems that are particularly sensitive to changes and in which maintaining ecological processes and services becomes challenging [55–57]. In this context, halting land degradation is a key issue for sustainable land management and planning, as worse environmental conditions increase land vulnerability and exposure to future natural hazards.

For these reasons, it is important to develop measures to contain land consumption and continuous monitoring of the phenomenon, especially in areas in which land consumption trends show an increase. Urban development affects future risks, and urban change analyses can improve our understanding of land cover/use change processes. This allows us, for example, to prioritize interventions and to consider areas with more relevant trends or with multiple impacts from urbanization, geomorphology, and hydrogeological hazard, providing a basis for informed decision-making for the purpose of sustainable territorial management. This is of strategic importance for Europe, which has a long coastline, including Italy, where many human activities are concentrated. Therefore, monitoring of changes and the assessment of specific vulnerabilities are fundamental for sustainable planning and effective policies and measures to reduce future exposure and to fulfil EU strategies and goals. Particular attention must be paid to coastal areas and the risk derived from land consumption, also considering the measures that will be established by the forthcoming EU Soil Health Law, which we hope will deal with the peculiarities of Mediterranean coastal areas and their fragile soils.

To conclude, it is necessary to strengthen the monitoring of changes along the coast and to support the development of comprehensive frameworks for the assessment of vulnerabilities, impacts, and risks, in order to better support territorial planning and restoration interventions, with particular attention to areas already significantly compromised.

**Author Contributions:** Conceptualization, D.S., A.C., C.G. and F.A.; methodology, D.S., A.C. and C.G.; formal analysis, A.C. and C.G.; writing—original draft preparation, D.S., A.C. and C.G.; writing—review and editing, D.S. and F.A.; supervision, F.A. All authors have read and agreed to the published version of the manuscript.

**Funding:** This research received no external funding.

**Data Availability Statement:** The data supporting the findings of this study are openly available at https://groupware.sinanet.isprambiente.it/uso-copertura-e-consumo-di-suolo/library/consumodi-suolo, (accessed on 10 January 2023). https://sinacloud.isprambiente.it/portal/apps/webappviewer/index.html?id=089e0739893f482e9e9b627360b6ff6d, (accessed on 31 January 2023). https://idrogeo.isprambiente.it/app/page/open-data, and http://esse1.mi.ingv.it/ (accessed on 31 January 2023).

**Conflicts of Interest:** The authors declare no conflict of interest.

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
