# Peer review of "The Increasing Coastal Urbanization in the Mediterranean Environment: The State of the Art in Italy"

_land, doi:10.3390/land12051017_

Round 1

Reviewer 1 Report (New Reviewer)

This paper provides an empirical study of coastal urbanization in Italy. The distribution of urbanized land is analyzed from socio-economic and geological perspectives, and implications regarding geo-hazard risks are given. 

This paper aims to raise the issue of potential problems caused by coastal urbanization. However it can yet be improved to better address this opinion by

1. Reorganize the narrative of the paper. The introduction is not sufficiently presented so that the aims and means of the paper can be clearly presented. 

2. Provide stronger and more complete evidence. The link between specific coastal urbanization in Italy and local geo-hazards is not complete. More elaborate explanations is expected.

English is Ok in general, but certain expressions can be polished for easier comprehension. A simple go-through by a native speaker or a proficient English user would be appreciated.

Author Response

Reviewer 2 Report (New Reviewer)

Please see document attached.

Manuscript is well written. English is appropriate. Just some long sentences need to be restructured and shortened.

Author Response

Reviewer 3 Report (New Reviewer)

The general approach and the motivations behind the study are very interesting. The quality of the data and maps is high. It appears as an interesting descriptive study.

However, a clear research structure is not appreciated. i.e.: There are no hypotheses that can or cannot be verified.

In general terms, it is recommended to rework the document towards a more investigative approach.

Other comments:

It is not explained why medium hydraulic hazard was selected (and not high or very high) (and the same question for the rest of the risks).

Figure 3. The degree of hazard could be added to the maps.

Figure 4. Consider contributing the digital maps with greater detail in the final version, in order to see more clearly the land consumption between 2012 and 2021.

Could some simple statistical analysis be done to show the relationship between the increase in land consumption and the population decline?

The relationship between hazard, vulnerability and risk is not properly defined.

The relationship between increased land consumption and increased risk is not sufficiently explored.

On the other hand, not all types of natural hazards (flood, landslide, seismic) interact in the same way with the vulnerability derived from greater urbanization.

Round 2

Reviewer 3 Report (New Reviewer)

The authors have responded to the suggestions in the cover letter, but have hardly applied them to the document. I still think they could incorporate some of the suggestions without a lot of work. The study deserves it and the readers would appreciate it.

Author Response

Thank you for your further revisions. Regarding Figure 3, the degree of hazards of the original maps is very detailed, and given the high fragmentation of the polygons, it is very difficult to understand the information at the scale of representation of the manuscript page. For this reason, we decided to merge the degree of hazards. For a detailed description of the degree of hazards, we provided the references of the source maps.

As for Figure 4, the detail of the information shown in the figure is the maximum possible taking into account the scale of representation of the manuscript page; to make the land consumption occurred between 2012 and 2021 more visibile we switched the colours of the maps (red for consumed land and black for land consumption). However, area changes are reported in the text and graphs in the “Results” section.

Regarding the other comments, the study presented in this manuscript is a co-occurrence analysis of coastal urbanization with other factors (coastal-inland gradient, natural hazards and population dynamics). To better address the causal relationship between urbanization and other factors, more analysis are required connecting information on drivers conditions in the different context. For example, areas prone to natural hazards represent a very general indicator that includes very different local situations whose relationship with urbanization must be studied in detail. Certainly, increasing urbanization in these areas will at least increase exposure, as it has already been mentioned in the manuscript in the “Introduction” and “Conclusion” sections. We will certainly take all valuable suggestions into consideration for future analyses, but including them in this manuscript would require a new set up of the whole work, with new objectives and new hypotheses to test. This would mean altering the rationale of this work in favor of a new approach, and this was not the aim of the paper. The shortcomings and possible improvements of the analysis have already been listed in the “Discussion” section.

This manuscript is a resubmission of an earlier submission. The following is a list of the peer review reports and author responses from that submission.

Round 1

Reviewer 1 Report

In their manuscript, the authors describe how the Italian coast is increasingly affected by urbanization and land consumption. For their analyses, they assessed the progressing transformation of land and its covering by artificial surfaces, in particular in six selected coastal municipalities. The authors relate their results to the dynamics of the human population and hydrogeological risks. This topic is of high relevance as land consumption poses a serious threat to ecosystem functioning (next to other negative consequences of dispersed urbanization). Linking settlement development with the risk of natural disasters is also very important, especially in the light of climate change. Therefore, I think that the manuscript addresses questions of high interest to a wider spectrum of scientific disciplines (planners, ecologists, human geographers ...).

However, I have had difficulties judging the scientific quality and novelty of the manuscript because, the authors do not describe their methods in sufficient detail. In the section “Reference Data”, the authors report to base their analyses on several available datasets and maps. This is alright, but in the section “Assessment of coastal areas changes”, I miss a detailed description on the methods or analytical steps they have applied themselves. How has the available data been processed? How was the amount of land consumption derived from the “national land consumption map of Italy”? Did the authors perform manual photointerpretation themselves, or had this been done by the authors of Strollo et al. 2020? Unfortunately, I could not open the link to the map provided in the manuscript – so I am not sure whether the amount of land consumption (i.g. the numeric values) could directly be retrieved from the map or whether this involved further analytical steps? The authors also refer to other datasets and maps: which information was retrieved from which source and how was the information combined? The same holds true for the maps of hydrogeological risks – the authors show three maps – but how did they select, exactly, the areas of interest for the analyses reported in this manuscript? And how were the borders of the coastal strips selected? Is there a reason for choosing 0-300, 300-1000 and 1000-10000 m distance from the coastline (from the introduction, one can refer that the fist strip is protected by national law, but what about the other two strips)? Does “land consumption of the six municipalities” refer to an area that falls into the first/second/third coastal strip? Or in other words: how does the area shown in Fig.3 relate to the area shown in Fig. 5?

All this is not stated in the very brief methods section of this manuscript and this makes it difficult to evaluate the results and conclusions reported by the authors.

Further comments:

- Also the information on performed calculations is very sparse: How was the “per capita consumed land” calculated, exactly: in total, per coastal strip, per municipality? (this can be inferred from the results but it should also be stated in the methods)

- Inconsistent use of the terminology: It should read “land consumption” throughout, not “soil consumption”

- Discussion: It is an interesting – and even alarming result – that land consumption takes place to quite an extent in areas that have a high risk of hazards. I think, elaborating on this issue, could increase the value of the manuscript substantially. So far, it is a bit difficult for the reader to judge, how problematic the additional land consumption in the affected area really is. Do problems arise, mainly, because the extra land consumption means more people in hazard-prone areas? Or does the extra land consumption increase the risk of further hazards? Or does it decrease the risk of certain hazards (e.g. does the extra land consumption stabilise or destabilise the areas and why?). The authors do refer to the problem of surface runoff (e.g. P10, L 258), but what about the other risks mentioned (landslide, seismic risks)?

Minor comments

- P5 L167: why “national consumed land”, when the numbers refer to the coastal strips?

- P5 L170: “allows us to understand” – this sound a bit strange, it is rather obvious that the striking difference between absolute and relative numbers results from the large differences in the sizes of the coastal strips. I would suggest highlighting the different size of the coastal strips in the methods section.

- P7 L209: Figure legend could be improved by stating that b) shows the consumed land for each coastal strip separately

Reviewer 2 Report

The manuscript entitled “The increasing vulnerability from urbanization of coastal areas: the case of Italy” aims to assess the effects of population dynamics (2012-2021) in terms of land cover changes and in relation to natural risks/hazard, focusing on coastal areas in Italy.

The paper puts forward an interesting topic, but the use of fundamental terms (i.e., risk, hazard, vulnerability) is wrong. I advise the authors to check the proper definitions of these terms (https://www.undrr.org/terminology), read them and properly use them. The research work deals with exposure caused by population increase and urbanisation, not with vulnerability. Vulnerability is more complex than exposure. Please see the official UNDRR definition: Vulnerability = "The conditions determined by physical, social, economic and environmental factors or processes which increase the susceptibility of an individual, a community, assets or systems to the impacts of hazards." (https://www.undrr.org/terminology/vulnerability). Therefore, the title must be changed.

I consider that this paper is not suitable for publication in Land, because it does not meet the standards of this ISI journal. Here are several of the reasons:

- The incorrect use of risk-related terms, which negatively affects the scientific information and causes confusion.

- The Introduction ends with the aim statement, but it should also discuss the implications of the findings both at national and international level.

- The Methodology section includes information that belong in Results (Figures 2, 3 for example). Also, this section does not give enough details about the methodological framework.

- In the Results section, all the data are reported to be for 2021. Where is the data for 2012? I think there is a mistake here. Check Table 1 and Figures 4, 5.

- The Discussion needs to be augmented to include comparisons with previous scientific findings specific to the study area/Italy, the implications of these results, the contribution of the paper to the field of scientific research, the limitations of the paper.

- Please note that there are some issues with the Figures’ captions too (Figure 1, 4, 5). Figures 2 and 3 should be part of Results.

I advise the authors to get a better understanding of what risk, hazard, vulnerability, exposure really mean according to international standards. Also, reading more papers on these topics (especially from high-rank journals), and analysing the structure of their sections should point the in the correct direction of what is expected from a manuscript intended for publication in ISI journals. In my opinion, this paper is just an early draft of a research work that is worth publishing, after a thorough reading of impactful scientific literature, reframing the concepts, thinking, rewriting, editing.

Reviewer 3 Report

The manuscript “The increasing vulnerability from urbanization of coastal areas: the case of Italy.” describes the state of the art of urbanization in Italian coastal area in 2021, both at national and regional levels.

Hence, thank you for considering the below mentioned remarks that address most of my concerns/issues regarding your article.

INTRODUCTION

Well-structured and written.

MATERIALS AND METHODS

Figure 2: Please verify the legends or clarify how you represent the various risks as a percentage. Is it possible to improve the resolution of the images? (Also for other figures)

RESULTS

Figure 4b. Friuli Venezia Giulia ad the F.

GENERAL COMMENTS

I believe that the work is a good contribution to understanding the critical issues related to degradation in the Italian coastal areas. The manuscript is presented in a proper form and for these reasons is needs only minor revisions.

I wish you good work!

Reviewer 4 Report

This paper describes the state of the art of urbanization in Italian coastal area in 2021, both 11 at national and regional level. Moreover, a focus was made on six coastal municipalities aiming to 12 evaluate the land consumption in relation to population dynamics between 2012 and 2021, and in 13 specific areas prone to natural risks along coastline. The research has certain practical value, but there are the following problems:

(1)         The research review is not enough, and the research progress in this field is not fully grasped.

(2)         The most important is the lack of innovation in this study.

(3)         The in-depth study of the relationship between land consumption and population dynamics is insufficient.

(4)         The discussion part of the paper does not put forward good policy suggestions.

(5)         Conclusion summary and refining are not enough and need to be strengthened.

(6)         Important literature should be supplemented in the literature section.